# Numerical simulation of the fine kinetics of dust reduction using high-speed aerosols

**Deji Jing**[1,2,3], **Jichuang Ma**[1,2,3], **Tian Zhang** [ID][1,2,3] *, **Shaocheng Ge**[4], **ShuaiShuai Ren**[1,2,3],
**Mingxing Ma**[1,2,3]

1 College of Safety Science and Engineering, Liaoning Technical University, Fuxin, China, 2 Research Institute of Safety Science and Engineering, Liaoning Technical University, Fuxin, China, 3 Thermodynamic Disasters and Control of Ministry of Education, Liaoning Technical University, Fuxin, China, 4 Safety and Emergency Management Engineering College, Taiyuan University of Technology, Taiyuan, China

☯ These authors contributed equally to this work.
* zhangtian@lntu.edu.cn

## Abstract

A numerical model of single-particle fog-dust collision coupling in a high-speed airflow based on three-phase flow theory. The effect of the fog-to-dust particle size ratio, relative velocity between the fog and dust particles, collision angle and contact angle at the wetting humidity function of dust particles is investigated. Different particle size ratios are determined for achieving the optimal wetting humidity for the interaction of high-velocity aerosols with dust particles of different sizes, for differ, that is, $k_{PM2.5} = 2:1$, $k_{PM10} = 3.5:1$ and $k_{PM20} = 1.5:1$. The optimal humidity increases with the relative velocity $U$ between the fog and dust particles in the high-speed airflow. The larger the collision angle is, the lower the wetting rate is. The smaller the contact angle between the solid and liquid is, the better droplet wetting on dust is. The fine kinetic mechanism of single-particle fog-dust collision-coupling in a high-speed airflow is elucidated in this study.

**Data Availability Statement:** The data comes from an unfinished project and cannot be made public for the time being. Data sharing rights are owned by Liaoning Technical University, please contact the data manager if you have any requests.E-mail: 904301642@qq.com.

## 1. Introduction

The most recent WHO report states that respiratory dusts with diameters of 10 μm or less (≤ PM2.5) are more harmful than ordinary dusts and that long-term exposure to these dusts increases the risks of cardiovascular disease, respiratory disease and lung cancer. Based on the WHO Air Quality Guidelines of an annual average maximum safe level of 10 μg/m³ or less for PM2.5, approximately 90% of the world's population is breathing polluted air [1]. Respiratory dust is a fine-particle pollutant with an aerodynamic diameter of 10 μm or less that is easily diffused and transported by wind flow and difficult to settle by gravity [2, 3]. The wet treatment methods, such as high-pressure spraying, dry fog-dust suppression and dust sprinkling, used to capture these particles have a low efficiency and a high cost, as well as poor applicability and stability [4]. Therefore, it is important to investigate efficient reduction methods for respirable dust to achieve daily clean production and improve natural environments to protect workers' health.

Gas–liquid two-phase spraying is an important fine-atomization spray technology used in the field of dust removal [5]. Compared with high-pressure atomization and ultrasonic

**Funding:** jdj-National Natural Science Foundation of China Youth Fund Project (51704146) jdj-Liaoning Provincial Natural Science Foundation (2020-MS-304) jdj-Liaoning provincial funding for scientific research projects (LJK0323) zt-China Postdoctoral Science Foundation (2022M11456) The funders had no role in study design, data collection and analysis, decision to publish, or preparation of the manuscript.

**Competing interests:** The authors have declared that no competing interests exist.

atomization, a gas–liquid two-phase spray has a higher power, larger spray volume, longer range, and wider coverage. Compared to other dust removal methods, a gas–liquid two-phase spray is less affected by wind flow disturbance and offers the advantage of forming a pool of a saturated water vapour mist at high evaporation rates to promote dust condensation [6]. Guang Wu et al. [7] carried out a force analysis and established theoretical equations for dust in gas-phase fluids and used the Euler–Lagrange method to construct a mathematical model for the migration of dust particles in an airflow at a fully mechanized working face, whereby laws for airflow migration and dust diffusion were determined under different ventilation conditions. Zhou Qun et al. [8] proposed a novel dust removal system for dust sources in a coal mining area, performed an in-depth study of the diffusion and fragmentation mechanism of liquid droplets in an atomisation field, and built a platform for dust reduction to measure the macroscopic atomisation performance of nozzles under different pressure conditions. Tian Zhang et al. used a transformation dimension method based on a CFD module. The supersonic atomisation process was simulated [9], and dust removal by a high-speed atomization water mist was experimentally verified [10]. Based on this principle, we designed and fabricated a standardised sprinkler with widespread application to industrial fields. This novel nozzle completely surpasses the traditional pneumatic atomization nozzle in terms of the droplet speed, conservation of water and pressure, range, and attenuation view. To elucidate the dust reduction mechanism, many researchers have used particle collision theory to analyse dust capture by gas–liquid two-phase atomisation and found that microdynamic interactions, such as collisions and wetting, between fine water droplets and dust, play an important role in dust capture. One of the representative results of these studies is the KIVA program, which was developed based on the particle collision statistical model [11]. These results have led many scholars to conclude that collisions of water mist with dust occur in a three-phase flow. In 2004, VM Alipchenkov et al. used mass, momentum and energy conservation equations for a gas phase, dispersed phase (droplets) and thin film to develop a three-fluid model of a two-phase dispersed annulus flow [12].

Haiming Yu et al. established a mathematical model of three-phase-flow based on the interactions between droplets, dust and an airflow and analysed the relationship between the spray pressure and dust reduction efficiency [13]. Qingtao Zhang et al. performed a LFNRM experiment in conjunction with micromolecular dynamic simulations to study the wetting relaxation times of lignite using different types of surfactants, as well as the wetting dynamics characteristics and action mechanism of lignite at the micro level [14]. Washino, K et al. used a combination of experiments and simulations to study the process of dust particles colliding with liquid droplets and adhering to a wall surface [15]. Sandip Pawar et al. experimentally investigated the law for interactions between droplets during spraying [16]. Xueming Fang et al. designed a test system to simulate spray dust reduction, evaluated the dust capture efficiency for a variety of water mists with different particle sizes and identified the main stages of dust capture [17]. Charinpanitkul T et al. investigated the characteristics of several complex factors associated with dust removal by a water spray. The effect of the nozzle type on the droplet size distribution and the effect of this distribution on the total dust removal efficiency were investigated by calculating the spray footprint area corresponding to the droplet trajectory [18]. Qingxin Ma et al. used a combination of spray experiments and numerical simulations to thoroughly investigate the spray characteristics and dust removal performance of external spray equipment used in coal mining. The particle size distribution was determined and used to evaluate the optimum dust reduction effect of the spray device [19].

Many domestic and international scholars have studied the relationship between the spray pressure, droplet size and dust reduction efficiency for the process of dust collection by a fine mist and determined the optimal spray pressure or droplet size (range). Under a specific

condition (the prescribed spray type, the spray code and a fixed contact angle between the solid and liquid surfaces), one droplet size is not suitable for all dust types. Given the enormous differences between the characteristics of and hazards associated with different dusts, different parameters are required for droplet collection of different dusts, such as PM2.5, PM10, and PM20; however, these parameters have not been determined in previous studies and the reasons for this behaviour have not been explored in depth. As previous studies have failed to determine either the optimal range of droplet sizes for the removal of typical dusts or the coupling mechanism under the combined impacts of the fog-to-dust particle size ratio, relative velocity between fog and dust particles, collision angle and contact angle at the solid–liquid surface, it is not possible to determine the optimal droplet size range for collection of monodisperse dust particles, which would be useful for the design and construction of spray systems.

Therefore, in this study, trinomial flow theory is used to build a numerical model of collision coupling between micron-sized high-speed droplet and dust particle. The line integral of the humidity function for the dust particle surface Φ is used to determine the effect of the various aforementioned factors on Φ. By analysing the competition between the inertial force and surface tension on wetting and the relaxation time under transient conditions, the mechanism for the collision coupling of fog and dust is elucidated to determine the optimal droplet size for the collection of different typical dusts. The clearly different optimal droplet sizes can be used to formulate guidelines for optimising the spray parameters and determining the droplet distribution of a spray system to improve the dust reduction efficiency.

## 2. Physical model of single-particle fog-dust collision-coupled three-phase flow in high-speed airflow

Supersonic atomisation is an advanced spraying technology with high efficiency and power. A liquid is atomised by mutual extrusion and accelerated shearing between two media, and the atomisation effect is enhanced by the airflow outside the nozzle. There is a large proportion of small- and medium-sized particles in the droplet size distribution, and the fog concentration increases exponentially with the water consumption, improving the wetting efficiency on coal dust. The breakup of a fixed volume of water into an infinite number of tiny droplets results in a significant increase in the surface area; the tiny droplets collide with fine dust particles in the air, and condensing and settling processes eventually realise the removal of the dust particles from air. During the process of dust reduction using a water mist, the efficiency of dust-particle capture by droplets is mainly affected by the inertial collision between particles and the contact wetting efficiency of the mist on the dust. The inertial collisions between the particles and the contact wetting efficiency of the mist on the dust are mainly affected by the fog-to-dust particle size ratio, relative velocity between the fog and dust, collision angle and contact angle at the solid–liquid surface, air velocity and the state of the flow-field of the dusty air stream.

Dust particles in air migrate and diffuse into a gas phase of droplets over time, and the dust particles inside the droplets undergo Brownian motion, resulting in a gradual decrease in the droplet volume percentage in air as the diffusion process continues, as shown in Fig 1. The internal structure of the droplets changes, where the largest phase in the droplets is liquid and known as the fog nucleus. Different droplet sizes and contact angles at the dust-droplet interface caused by the physical and chemical properties of the collision surface manifest as different wetting properties. These different wetting properties result in the formation of a gas-liquid bridge at the gas-liquid-solid three-phase interface. The contact angle at the boundary affects the linkage between the fog nucleus and the dust particle, where different wetting angles result in the formation of fog bridges with different widths, lengths, and depths. As a result,

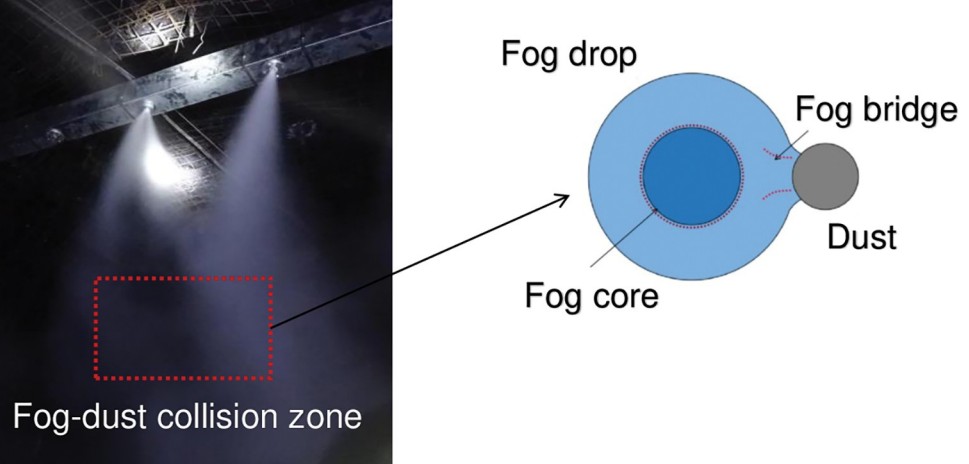

**Fig 1. Schematic of fog wetting a dust particle during collision.**

the transient wetting rate of the droplets on the dust particles can be determined. The imbalance between the two phases diminishes over time and is referred to as "slack". In a droplet-particle flow, the main equilibrium is velocity equilibrium, and the dust particles are wetted by the liquid droplets in this dynamic state.

Fig 2 shows the geometric model used to numerically study the coupled three-phase flow in which single-particle fog-dust collisions occur in a high-speed air stream. The figure shows the two colliding objects, i.e., the fog droplets and dust, as well as the ambient air domain in which the collisions occur, i.e., the figure shows the high-speed air flow, fog droplet and dust particle. In this study, dust particles with diameters of 20 μm, 10 μm and 2.5 μm are considered, and

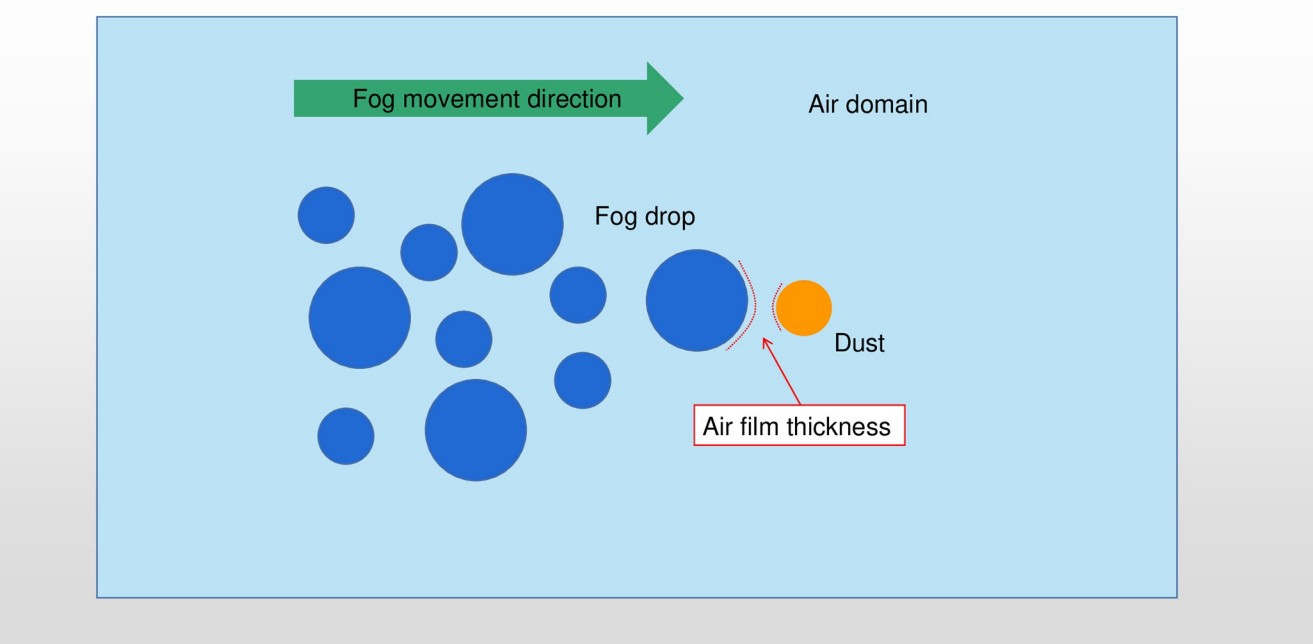

**Fig 2. Geometric model of dust-fog collisions.**

the droplet size is set accordingly; the relative velocity of the droplet to the air domain is 0, and the relative velocity of the dust particle to the droplet is set accordingly. The air domain is a square with sides 200 times larger than the diameter of the particles.

## 3. Numerical simulation of single-particle fog-dust collision-coupled three-phase flow in high-speed airflow

Dust reduction by pneumatic atomisation involves the production of a high-velocity aerosol; thus, a numerical model with a high-velocity airflow as the background is constructed. Unlike the fog-dust coupling process in a high-speed airflow, in a conventional high-pressure spray-dust coupling, there is relative motion between the high-pressure spray droplets and external air, as well as between the dust and external air; the ambient air is flowing at a low speed when the dust is trapped and therefore has a weak effect on the relative velocity between the fog and dust [10]. A high-speed airflow for the fog-dust coupling process is realised using a high-speed pneumatic two-fluid microatomisation nozzle; thus, the combination of an outward spray and dust produces small droplets with low inertia and almost the same speed as the high-pressure, high-speed airflow to form a "high-speed aerosol", resulting in a large relative velocity between the aerosol and the dust [9].

In this study, we investigate the effect of the fog-to-dust particle size ratio, relative velocity between the fog and dust, collision angle and contact angle on the fine kinetic behaviour of dust-dust coupling; in addition, we elucidate the wetting mechanism during capture of dust particles during instantaneous collisions and the relaxation of the system to uniform motion after contact between the high-speed aerosol and dust.

### 3.1 Mathematical models

The greatest obstacles to the numerical simulation of gas-liquid-solid multiphase flows are the uncertainty in the existence, deformation and location of the gas-liquid interface and the rapid changes in the dynamics of the interface when the gas-liquid-solid phase interacts.

The level-set method was originally proposed by Osher and Sethian (1988, 1996) and was initially applied in the fields of intelligent image control and image processing before being adapted by Sethian and Sussman et al. for use in numerical simulation studies of gas-liquid two-phase flows. The basic idea of the method is to propagate the gas-liquid interface with a higher-order function level set, $\varphi$, with the zero value represented; the different phases in the computational region are distinguished by the algebraic value of $\varphi$. When combined with the finite element dynamic mesh technique [20], the level-set method can solve the flow, fragmentation and fusion problems at the gas-liquid-solid interface, where surface tension is the main physical parameter of the process. Surface tension acts as an energy at the phase interface and is mainly determined by the surface curvature and the normal vector [21].

To determine the kinematic interface, two transport equations [22], Eqs (1) and (2), which include the phase field variable $\varphi$, the mixing energy density $\psi$, the surface tension coefficient $\sigma$, and the interface thickness parameter $\varepsilon_{pf}$, must be solved:

$$\frac{\partial \varphi}{\partial t} + u\nabla\varphi = \nabla \cdot \frac{\gamma_0}{\varepsilon_{pf}^2}\nabla\psi \tag{1}$$

$$\psi = -\nabla \cdot \varepsilon_{pf}^2 \nabla\varphi + (\varphi^2 - 1)\varphi + \left(\frac{\varepsilon_{pf}^2}{\lambda}\right)\frac{\partial f}{\partial \varphi} \tag{2}$$

where $u$ is the velocity of motion, m/s; $\lambda$ is the surface energy density, J/m$^2$; $\gamma_0$ is the mass

mobility, $m^3 \cdot s/kg$; $\lambda$ and $\gamma_0$ are determined by Eqs (3) and (4), respectively; $\varepsilon_{pf}$ is the interface thickness parameter, m, which defaults to half of the maximum grid element $h_{Max}$ in the computational domain, i.e., $\varepsilon_{pf} = pf.h_{Max}/2$; and $\partial f/\partial \varphi$ denotes the field derivative of the external free energy f, $J/m^3$.

$$\lambda = \frac{3\varepsilon_{pf}\sigma}{\sqrt{8}} \qquad (3)$$

where $\sigma$ is the surface tension coefficient, N/m.

$$\gamma_0 = \chi \varepsilon_{pf}^{\,2} \qquad (4)$$

where $\chi$ is the mobility adjustment factor [22], m·s/kg, which determines the time scale of Cahn-Hilliard diffusion and needs to be kept large enough to maintain a constant thickness at the interface. However, too large a value allows excessive phase diffusion in the simulation, while too small a value produces diffusion damping, impeding the phase diffusion motion, which can be defined by Eq (5).

$$\chi = \frac{\boldsymbol{u} \cdot h_{max}}{3\sqrt{2}\sigma\varepsilon_{pf}} \qquad (5)$$

The transfer of the phase-field variables is accomplished by the convection of the velocity field $u$. The location of the fluid-fluid interface is determined by phase initialisation during the calculation. The gas-liquid interface is determined by the nature of the surface tension between the gas and the solid, which is included in solving the momentum equation. In addition, the liquid-solid interface is defined by the wetting wall, with a fluid-solid contact angle of $\theta_w$, rad. The surface tension term at the phase interface is defined as:

$$n \cdot \varepsilon_{pf}^{\,2}\nabla\varphi = \varepsilon_{pf}^{\,2}\cos(\theta_w) \cdot \varphi|\nabla\varphi| \qquad (6)$$

where n denotes the intrinsic geometric characteristic parameter at the interface, the normal vector. The mass through the fluid interface is 0, which is expressed as:

$$n \cdot \frac{\gamma_0\lambda}{\varepsilon_{pf}^{\,2}}\nabla\psi = 0 \qquad (7)$$

where the contact angle can be defined by Young's equation as:

$$\sigma\cos(\theta_w) + \gamma s2 = \gamma s1 \qquad (8)$$

where $\gamma_{s1}$ is the surface energy density at the interface between fluid 1 and the solid, $J/m^3$, and $\gamma_{s2}$ is the surface energy density at the interface between fluid 2 and the solid, $J/m^3$.

## 3.2 Degree of wetness

The wetting process after the collision of droplets with dust particles is not an instantaneous behaviour, but a continuous process, and the wetting process is influenced by a number of factors. Factors such as fog-to-dust particle size ratio, relative velocity, collision angle and solid-liquid surface contact angle. Since the instantaneous volume fraction $\varphi_l$ of the liquid phase at each point on the dust particle surface is not the same, to characterize the instantaneous wetting degree of the droplets on the dust particles, a definite integral function was defined on the circumference of dust particles to obtain the wetting humidity Φ.The degree of wetness Φ, μm, is defined by the line integral $\int_a^{a'} \varphi_l(x)dx$, as shown in Fig 3.

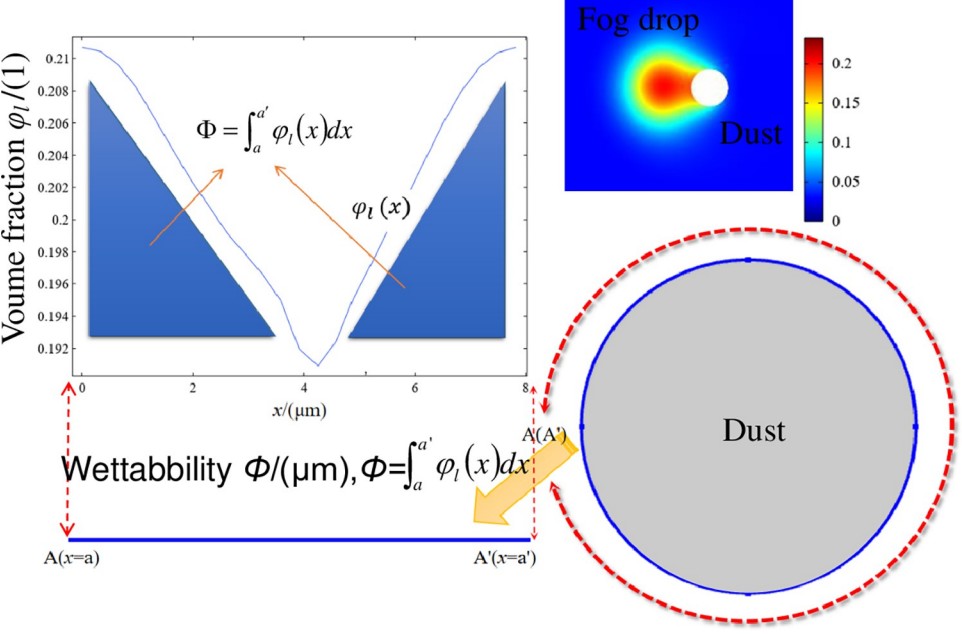

**Fig 3. Illustration of wetting line integral definition.**

### 3.3 Grid division

The computational mesh was classified by the hydrodynamic free triangular polarization of both the gas and liquid parts of the fluid, with a maximum cell size of 1.34 μm, a minimum cell size of 0.004 μm, a maximum cell growth rate of 1.05, a curvature factor of 0.2, and a narrow region resolution of 1. The solid region was classified by ordinary physics-free triangular meshing. The grid cell number was 66776, the minimum cell mass was 0.5547, the average cell mass was 0.9426, the cell area ratio was 0.01074, and the grid area was 40,000.0 μm². The boundary capture technique for a hydrodynamic polarization dynamic grid was used, with the gas and liquid phase regions set up as dynamic grids capable of capturing the transient boundary morphology changes of the gas and liquid phases at the microsecond level during the high-speed collision process. The grid of the gas-solid-liquid junction region is shown in Fig 4.

Fig 4 shows that the grid clearly delineates the droplet and dust particle boundaries and that the grid quality is good in the region of the moving grid where the droplet and dust particle meet. The morphological changes in the droplets and air film can be captured at high relative velocities.

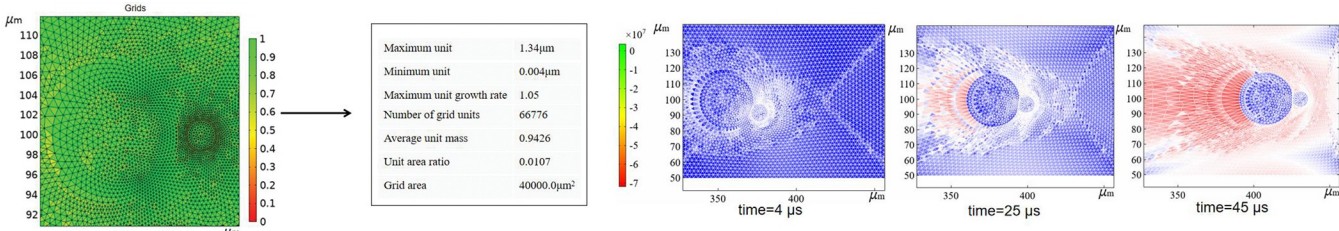

**Fig 4. Graphical grid of the gas-solid-liquid three-phase junction region.** (a) Grids and parameter setting, (b) Dust and droplet particle collision dynamic grid variation diagram.

**Table 1. Establishment of boundary conditions.**

| Boundary conditions | Set values |
|---|---|
| Dust particle density/kg·m$^{-3}$ | $1.33 \times 10^3$ |
| Young's modulus of dust particles/MPa | 2713 |
| Poisson's ratio of dust particles | 0.339 |
| Gas phase density/kg·m$^{-3}$ | 1.27 |

The boundary conditions for the numerical simulation are shown in Table 1.

## 3.4 Reliability verification analysis

Research on three-phase flow is primarily focused on the study of bubbles, oil, water and solids as media, and there is little research on the study of droplet-dust particle collisions. The existing literature is still lacking, especially that on studies that modelled collisions between micron-sized particle droplets in airflow at high relative velocities; there have been no relevant collision behaviour characterization and visualization experimental studies.

As a result, the results of the simulation "Distribution of the flow lines of the gas phase outside the dust particles after the transient collision of the two phases of fog and dust" were compared with the experimental results of the literature "Distribution of flow lines of an incompressible fluid when moving around a smooth cylinder". The results of the simulations were compared with the experimental results of "Variation of stress distribution in dust-mist collisions at different angles" and "Variation of instantaneous stress coefficients and coefficients of combined forces when incompressible fluid impact bypasses a cylindrical surface". The simulation was found to be reliable in terms of flow distribution and stress variation [23], and the morphology agreed well with published results of incompressible fluid flowing around a smooth cylinder without separation.

As shown in Fig 5, a hydrophobic case with a wetting angle of 180˚ was investigated in a collision wetting study with the contact angle as a variable, and the reliability of the simulation was confirmed by the presence of hydrophobic wetting and shielding phenomena. The wetting angle of 180˚ showed that the hydrophobic surface was not wetted by the droplets, and the simulation accurately depicted this insulating shielding phenomenon.

The fog-dust transient collision body streamline distribution state can be clearly seen in Fig 6. The fog droplets collide with the dust, and a thin air film forms in the middle; as the collision proceeds, the air is gradually squeezed out, and the particles in the solid flow around the air movement. The spacing between the airflow streamlines at the contact interface initially expands and then begins to decrease as the two enter a relaxed state. The spacing between the trailing streamlines of the dust particles initially decreases and then gradually increases after relaxation, which is consistent with the experimental results of "Distribution of incompressible fluid streamlines around smooth cylindrical surfaces" [24].

As shown in Fig 7, for the three dusts PM2.5, PM10 and PM20, the smaller the contact angle is, the better the wettability. When the contact angle is small, the wetting rate of the liquid phase to the solid phase is large, which results in high traction stress between the phases, making the droplets more likely to push the dust towards the nucleus and form a wetting fog bridge. When the contact angle is π, unlike in the previous case, the wetting rate after contact gradually decreases. The reason the contact angle is not 0 is that the droplet and dust particle always adhere together due to the traction of the airflow, even in the absence of wetting, and the volume fraction of the adjoining liquid phase is calculated by default statistics. This wetting/shielding phenomenon [25] also contributes to the reliability of the simulation process.

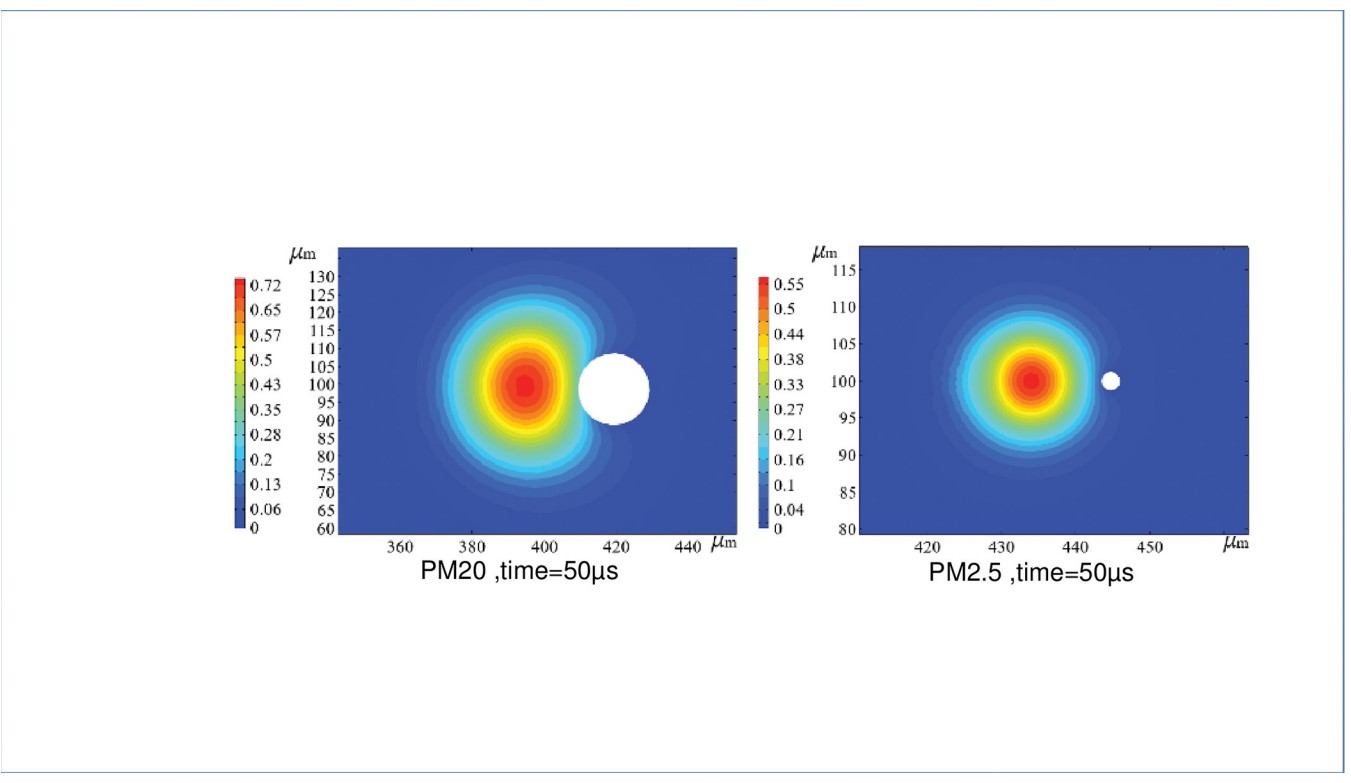

**Fig 5. Fog-dust transient collision shielding phenomena.**

## 4. Numerical simulation results and discussion

### 4.1 Optimal wetting mist-dust particle size ratios for different particle sizes

The particle size ratio of fog droplets to dust [26] is an important factor in the study of dust-dust coupling characteristics in the aerosol dust reduction process because it establishes a numerical model of aerosol-dust coupled-collisional three-phase flow. The collision coupling process was simulated for three different dust particle sizes with varying fog-dust particle size ratios $k$. In the simulation, the three dust particle sizes were PM2.5, PM10 and PM20, the relative velocity of the aerosol dust was 50 m/s, and the wetting angle was $\pi/9$.

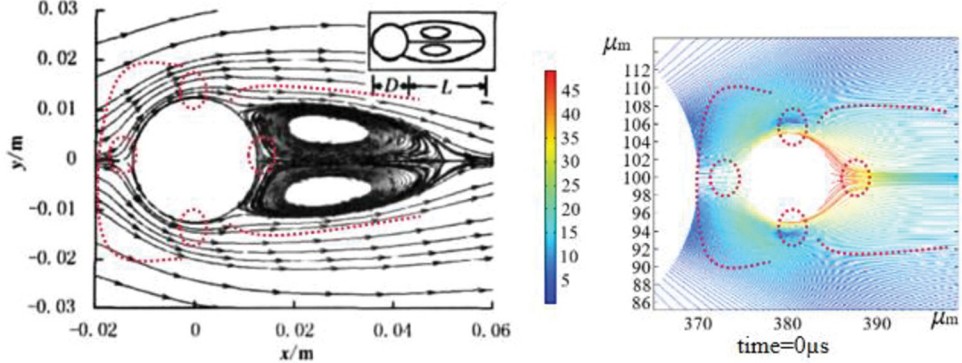

**Fig 6. Flow line distribution of fog-dust instantaneous colliding bodies.**

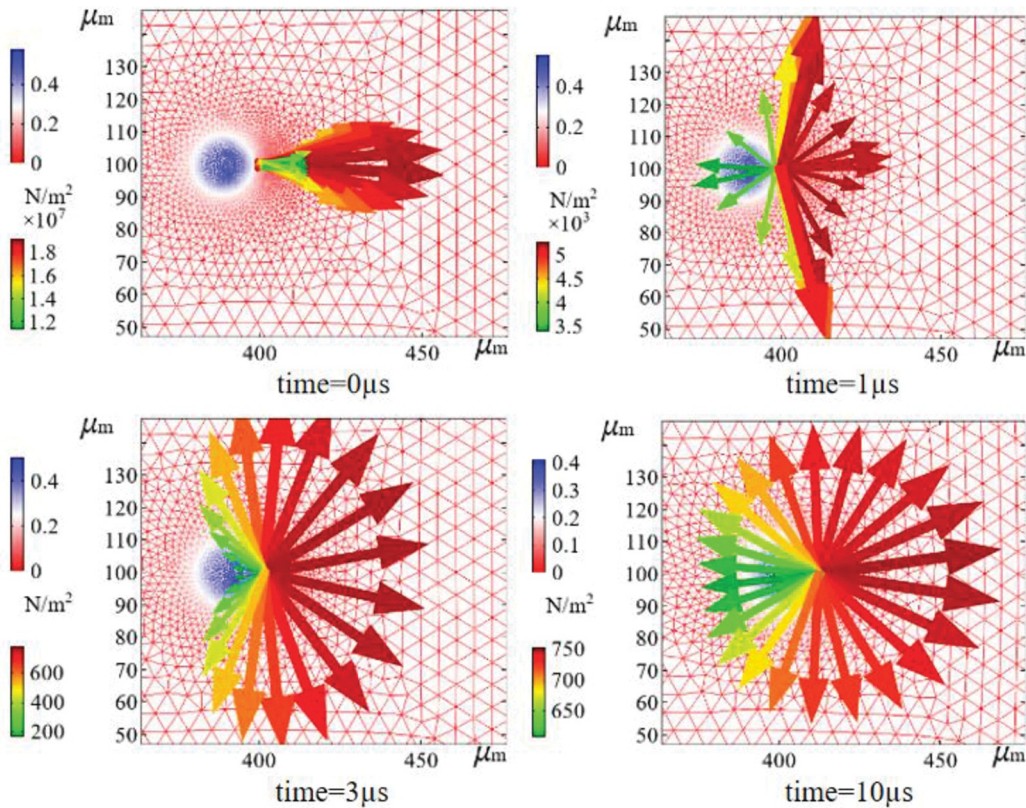

**Fig 7. Transient stress distribution of dust at different fog-dust collision angles.**

As seen in Fig 8, for the three dust particle sizes, the larger the relative velocity is at the beginning, the larger the relative velocity overall, resulting in different degrees of deformation dominated by droplet-dust inertia. The larger the dust inertia is, the greater the deformation at the same fog-to-dust particle size ratio, resulting in different particle sizes relative to the different dust sizes and different particle size ratios. In addition, the smaller the dust particles, the later the fog bridge is established, and the weaker the intensity. When the dust particle size is 2.5 μm, the optimal dust capture ratio is $k = 2:1$; when the dust particle size is 10 μm, the optimal ratio $k = 3.5:1$; and when the dust particle size is 20 μm, the optimal ratio $k = 1.5:1$. The optimal particle sizes for PM2.5, PM10 and PM20 are 5 μm, 35 μm and 30 μm, respectively, in high-speed aerosol dust reduction. This explains the low efficiency of conventional pneumatic fogging nozzles for dust below PM10, because the particle size distribution of commonly-used fogging nozzles is mostly in the range of 30 to 1000 μm, which is clearly not in line with the optimum particle size ratio.

## 4.2 Effect of the relative velocity of dust particles on collision-coupled wetting

The fog-dust transient collision body streamline distribution state clearly shows that, before the fog and dust collided and formed a crescent-shaped airflow that flowed backwards along the upper and lower surface of the dust particles, the dust particle movement in the opposite direction of the surface airflow first converged at a high speed to form a vortex and then collided. At 1 μs, the relative motion velocity decreased to 1.5~2 m/s. At 2 μs, the relative motion

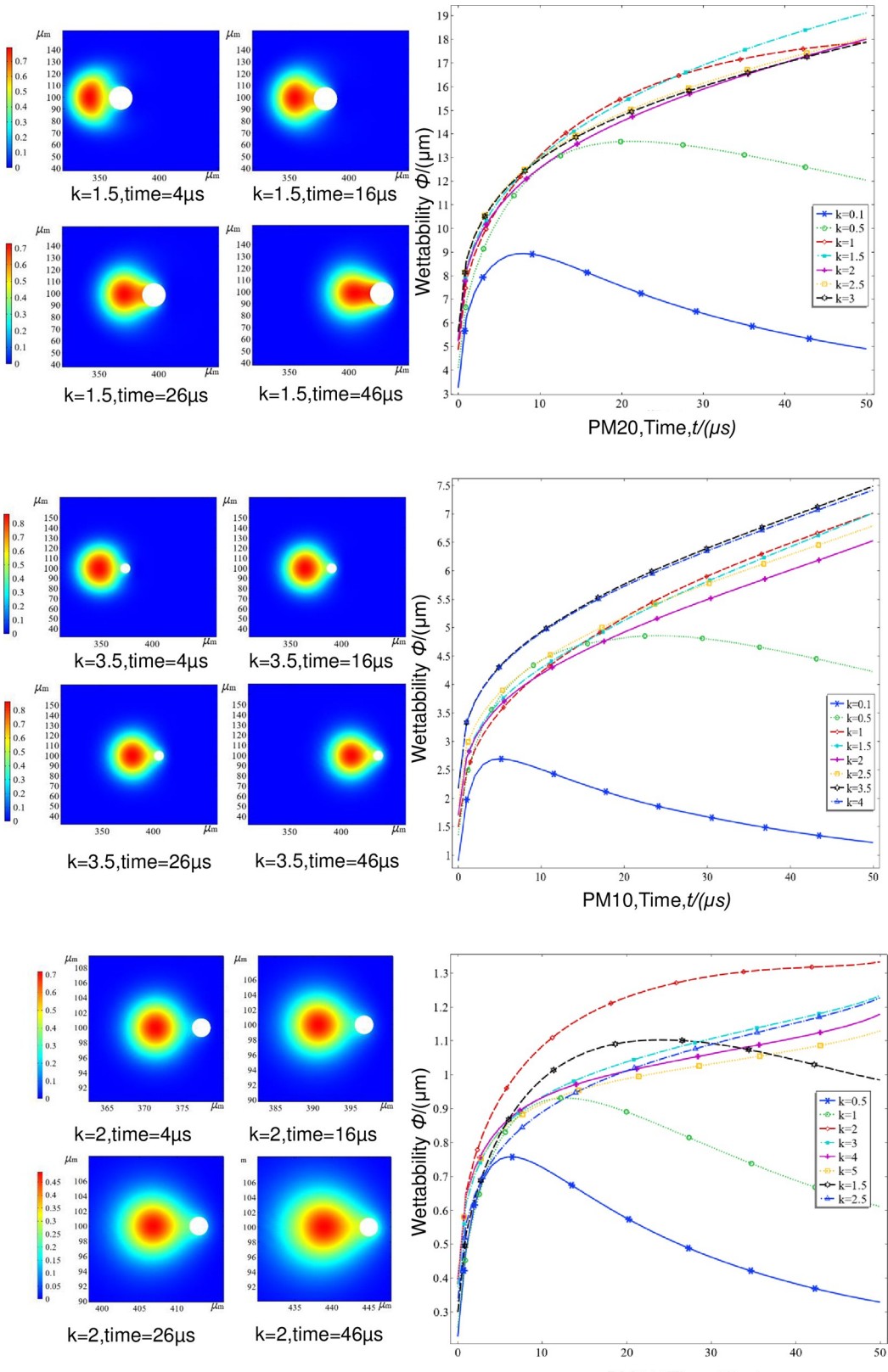

**Fig 8. Transient wet moisture of fog-dust collisions with different *k* values.** (a) Graph of transient collision data for PM20, (b) Graph of transient collision data for PM10, (c) Graph of transient collision data for PM2.5.

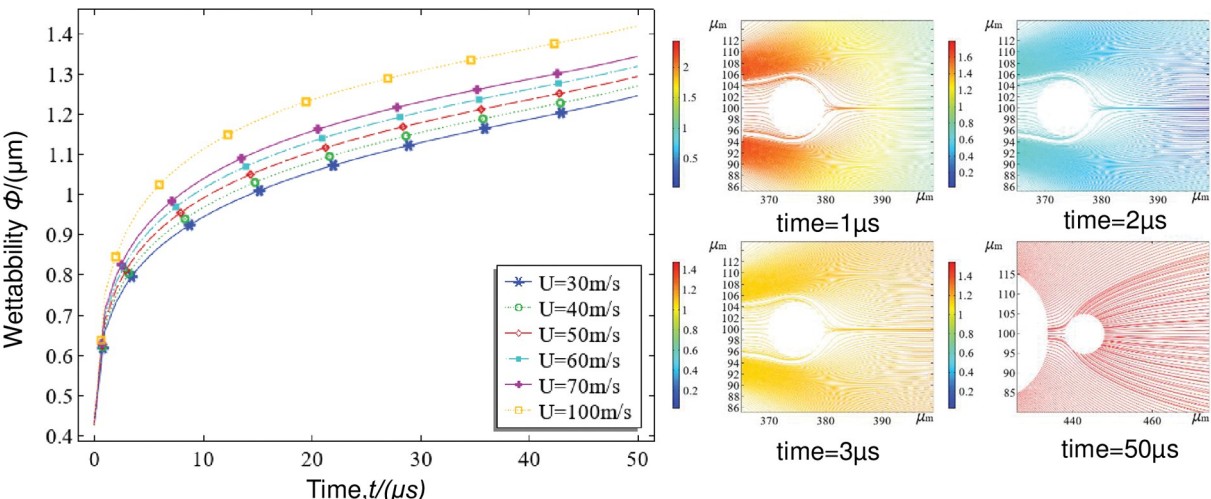

**Fig 9. Transient comparison of the wetness at different fog-dust relative velocities.**

velocity decreased to 0.4~0.6 m/s. When the velocity decreases to 0.2 m/s or tended to be relatively static, the velocity of the droplet and dust particle began to increase under the action of the airflow, with the velocity of the airflow on the right side of the dust particles being greater than the velocity of the airflow on the left side. At 5 μs, the front and rear velocities increased to 1 m/s at the same time, which occurred when the dust particles and droplets entered relaxed states simultaneously and were relatively stationary in the airflow, advancing at a uniform speed. From 8 to 50 μs, the dust particles were gradually brought closer to the droplets due to the thrust of the droplets, the traction of the airflow and the wetting traction of the droplet surface tension, and the streamline density between the dust particles increased and decreased in the forwards direction of the dust particles.

Fig 9 shows that the wetting rate increases with time for different fog-dust relative velocities, but as the relative velocity $U$ increases from 30 m/s to 100 m/s, the wetting rate decreases and the wetting value increases from 1.23 μm to 1.42 μm at 50 μs. It can be concluded that the wetting rate increases as the fog-dust relative velocity $U$ increases.

## 4.3 Effect of fog-dust particle collision angle on collision-coupled wetting

First, c is defined as the calculated adjustment factor for each series in the simulation, and the angle of particle collision is defined as $ß = c·π$ in terms of the initial relative velocity. For example, when $c = 0$, the droplets collide head-on with the dust particles. Fig 10 shows a graph of the wetting humidity variation from 1 to 10 μs for $c = 0.25$, $c = 0.16667$, $c = 0.08333$ and $c = 0$. The four angles of wettability are consistent. The wettability increases rapidly after the collision, and then the rate of increase gradually decreases due to the dehumidification effect, reaching a maximum value of 1.3 μm. The larger the angle of collision is, the lower the instantaneous wettability. The reason for the above phenomenon is that the interphase traction stress before the fog-dust relaxation movement varies depending on the fog-dust impact angle.

When the droplets collide with the dust particles, the dust particles are first subjected to the combined force of the opposite angle of impact, which is the surface tension of the liquid-phase surface wetting traction. As the impact progresses, the reaction force weakens, the traction force along the impact direction normal to both sides of the extension decreases, and the stress gradually decreases, The normal and impact direction for the opposite side of the force are greater than for the same side, because as the droplet and dust particle enter the relaxation

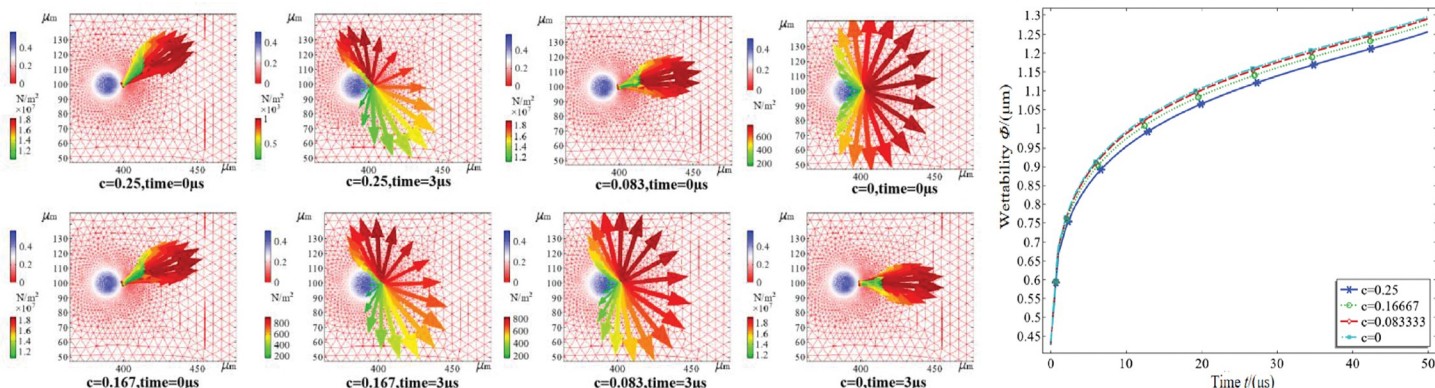

**Fig 10. Variation of wetness at different impact angles.** (a) Transient stress distribution of dust at different fog-dust collision angles, (b) Transient wetting diagram for dust at different contact angles.

state, the upper and lower sides of the force gradually balance. Due to the thrust of the droplets on the dust particles, the traction of the airflow on the dust particles is greater than the traction adsorption force caused by the wetting of the liquid-phase surface tension. The larger the impact angle is, the longer the time it takes to enter the relaxation state, the lower the rate of wetting, and the worse the effect of instantaneous wetting.To sum up, The competition between the wetting force of the droplets on the dust, the collision inertial force and the liquid cohesion reaction force on the fog-dust coupling characteristics differs for different collision angles; the larger the collision angle is, the lower the wetting rate, the longer the relaxation time and the worse the wetting effect.

## 4.4 Effect of the contact angle at the fog-dust interface on collision-coupled wetting

The solid-liquid contact angle [27] is an important factor in the wetting of solid dust particles by droplets and is often determined by the physical and chemical properties of the solid and liquid. For water, the solid surface is hydrophilic if the contact angle with the solid is less than $\pi/2$ and hydrophobic if it is greater than $\pi/2$. In this paper, the fine kinetic characteristics of the wettability of aerosol-dust coupling were investigated for three particle sizes and 12 contact angles with forward collisions at a relative velocity of 50 m/s. The contact angle (wetting angle) was defined in the simulations as $\Theta_w = \pi/c$, and the magnitude of the contact angle was varied by adjusting the value of the coefficient c.

As shown in Fig 11. The smaller the contact angle between the liquid and solid is, the better the wettability of droplets under the same conditions. This is reflected in the long traction distance between the droplets and the dust, the long, wide fog bridge and the rapid decrease in the volume fraction of the fog nucleus. When these factors are combined with the above analysis of the stress distribution between fog and dust, it can be seen that for PM10, the smaller the wetting angle is, the greater the traction between the fog and dust, the faster the relaxation rate, the flatter the droplet shape, the more complete the adsorption/wrapping of dust, and the better the wettability. For PM20, at 50 μs, because the inertia of the dust particles is higher than that of PM10, when a wetting shield is used, PM20 squeezes the droplets with a larger deformation, almost half-embedded inside the droplets, and there is no wetting behaviour observed in the dust particles. When $c = 9$, the fog bridge, fog nucleus and dust are all connected, and the wetting effect is improved. When $c = 18$, for near complete wetting, the mass-migration range of the droplets in the air wraps the dust completely.

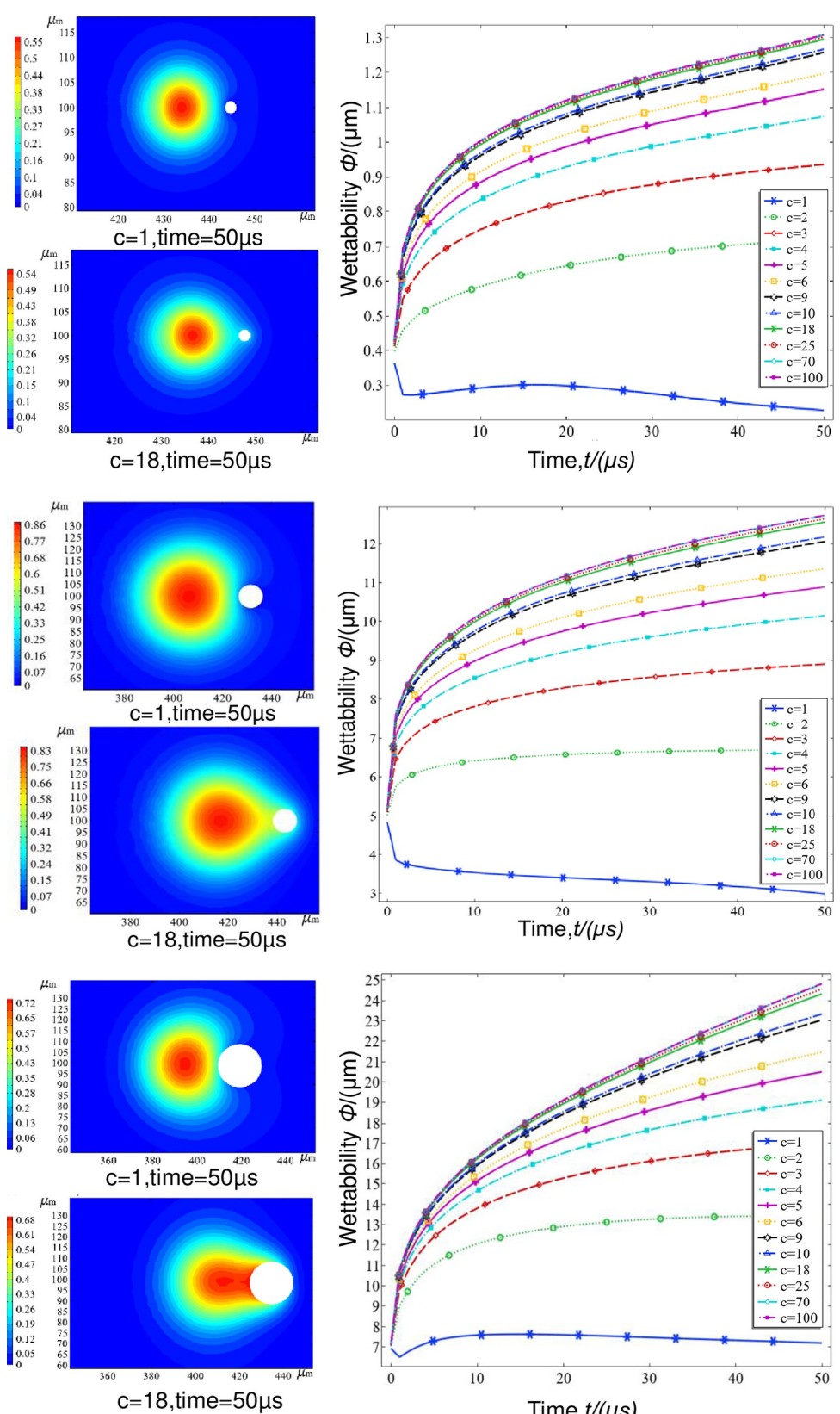

**Fig 11. Transient wetness of three kinds of dust with different contact angles.** (a) Collision diagram for different contact angles at PM2.5, (b) Collision diagram for different contact angles at PM10, (c) Collision diagram for different contact angles at PM20.

In contrast, PM2.5, due to its small mass, low momentum and poor inertia, is only able to cause minor deformations in the droplets and is not completely wrapped by the mass-migration range of the droplets in the air, despite being in close contact with the droplets when shielded. However, at small contact angles, the surface traction generated by the wetting action is greater and rapidly wets the dust, but due to the presence of a surface air film, the dust is not able to enter the interior of the fog nucleus until 50 μs, despite being completely wrapped by the penetrated area. The smaller the contact angle between the liquid and solid is, the better the wetting of droplets under the same conditions. This is reflected in the long traction distance between the droplet and the dust, the long, wide fog bridge and the rapid decrease in the volume fraction of the fog nucleus.

## 5. Experimental scheme

### 5.1 Experimental setup

Fig 12 shows the experimental platform that was set up to conduct experiments on three-phase gas-liquid-dust coupling. The main body of the dust reduction box was a cylindrical enclosed structure with a height of 150 cm and a diameter of 25 cm. Two types of dust-reduction nozzles were placed in openings directly above the box: ultrasonic atomising nozzles (which produced a water mist with a droplet size range of 25–60 μm) and supersonic atomising nozzles (which produced a water mist with a droplet size range of 8–35 μm). The dust reduction box was equipped with a dust-generating device and a dust-sampling device with symmetrical openings placed 50 cm above the bottom of the box.

The most accurate method for sampling, drying, and weighing dust [28] was used to measure the dust concentration, and a dryer tail sampler filter membrane was used to prevent the spray and airborne moisture from affecting the measurement results for the dust quality. The dust concentration at the test position was used instead of the average dust concentration in the chamber. The coal used in the experiment was coal dust ground from lumpy bright coal in the Fuxin area of Liaoning Province. The particle size distribution of the coal dust was as follows: 2.5 μm (5.7%), 2.5–20 μm (61.6%), 20–30 μm (31.1%) and more than 30 μm (2.6%).

### 5.2 Experimental procedure

An experiment is performed to determine the effect of the dust-to-mist particle size ratio on the dust reduction efficiency. First, a supersonic atomising nozzle is installed, the total mass of the dust generated by the dust collector is determined by weighing, and the pneumatic pressure is adjusted to 0.6 MPa to ensure uniform dust generation. The initial concentration of the dust generated in the chamber is measured. The spray is turned on, and the dust mass in the box is sampled at uniform time intervals; the dust is dried and weighed, the average dust reduction efficiency over the sampling time period is calculated, and the time period is replaced with the median moment to determine the average dust concentration at a given moment. The sampler flow rate is 20 L/min, the water flow rate is 80 ml/min, and the pneumatic total pressure of the nozzle is 0.4 MPa. The sampling filter membrane is dissolved, made into slides, viewed using a microscope observer, and the dust dispersion is calculated. The supersonic atomisation nozzle is replaced with an ultrasonic atomisation nozzle, the experimental steps are repeated, and the results are compared to those obtained for the previous test.

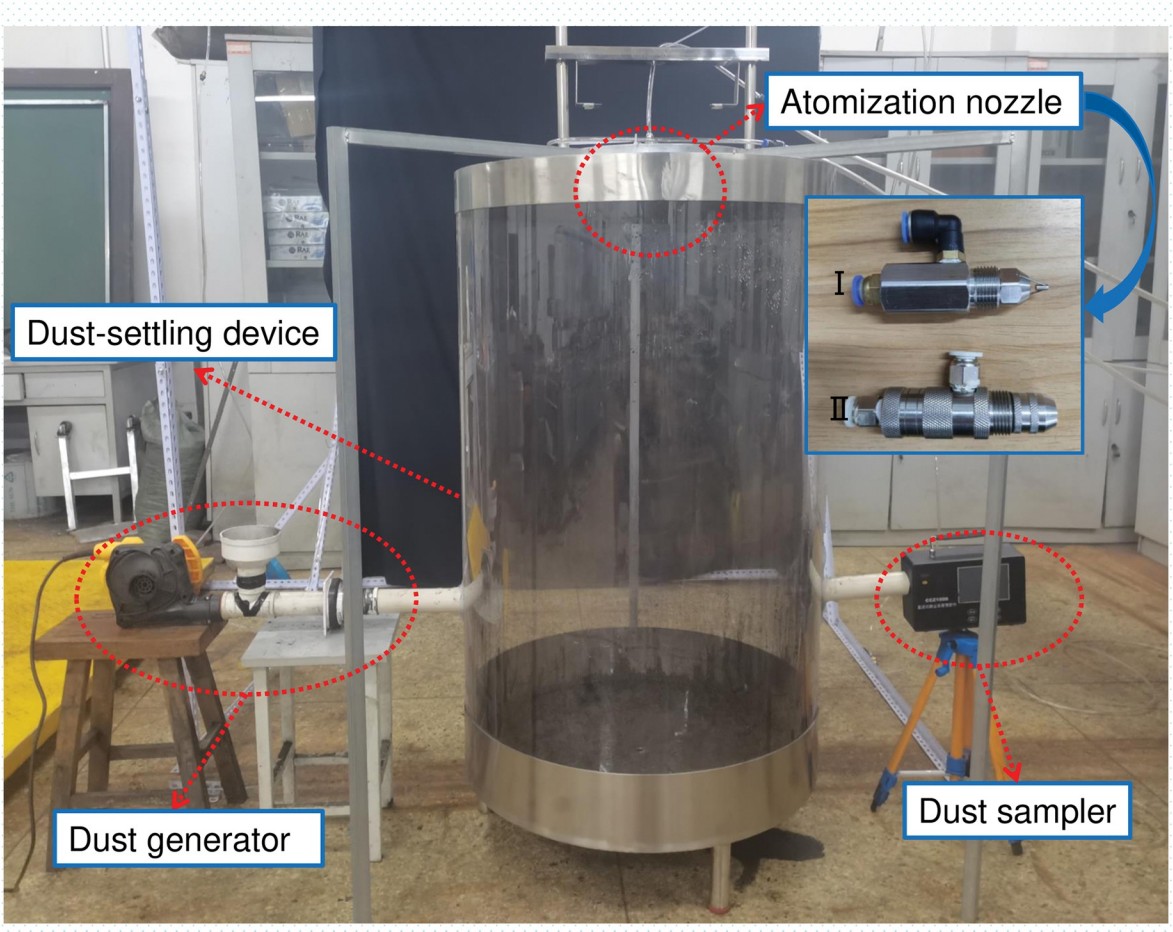

**Fig 12. The experimental platform used to generate a three-phase flow (gas-liquid-dust).**

Experiments are carried out to determine the effect of the relative velocity between the fog and dust particles on the dust reduction efficiency: the positions of the dust sampler and the materialisation device are first switched to produce a head-on collision between the streams of the dust particles and water mist. The supersonic atomisation nozzle is installed, the total mass of dust generated by the dust collector is determined by weighing, and the pneumatic pressure is adjusted to 0.6 MPa to ensure uniform dust generation. The initial concentration of the dust generated in the chamber is measured. The spray is turned on, and the dust mass in the box is sampled at uniform intervals; the dust is dried and weighed, and the average dust reduction efficiency over the sampling time period is calculated; the sampling time period is replaced with the median moment to obtain the average dust concentration at a given moment. The sampler flow rate is 20 L/min, the water flow rate is 80 ml/min, and the pneumatic total pressure of the nozzle is 0.4 MPa. The sampling filter membrane is dissolved, made into slides, viewed with a microscope observer, and the dust dispersion is calculated. The pneumatic pressure for dust generation is adjusted to 0.3 MPa, the experimental steps were repeated, and the results are compared to those of the previous tests.

## 5.3 Experimental results and analysis

Fig 13 shows the results of the experiments performed to determine the effect of the dust-to-mist particle size ratio on the dust reduction efficiency. Water mist particles of different sizes have different capture abilities for dust particles of a fixed size. The final dust reduction efficiency is 98.47% for the supersonic atomisation nozzle and 96.5% for the ultrasonic atomisation nozzle; the dust reduction rate of the supersonic atomisation nozzle is higher than that of the ultrasonic atomisation nozzle. The numerical simulation results show that the supersonic atomisation nozzle generates a water mist with a more suitable particle size for capturing the experimental dust than the ultrasonic atomisation nozzle, resulting in more effective dust reduction. The pressure of the dust-generating device is varied to change the velocity of the dust particles, and Fig 14 shows that the relative velocity between the dust particles and the mist changes with the dust reduction efficiency. When the dust-generating pressure is reduced

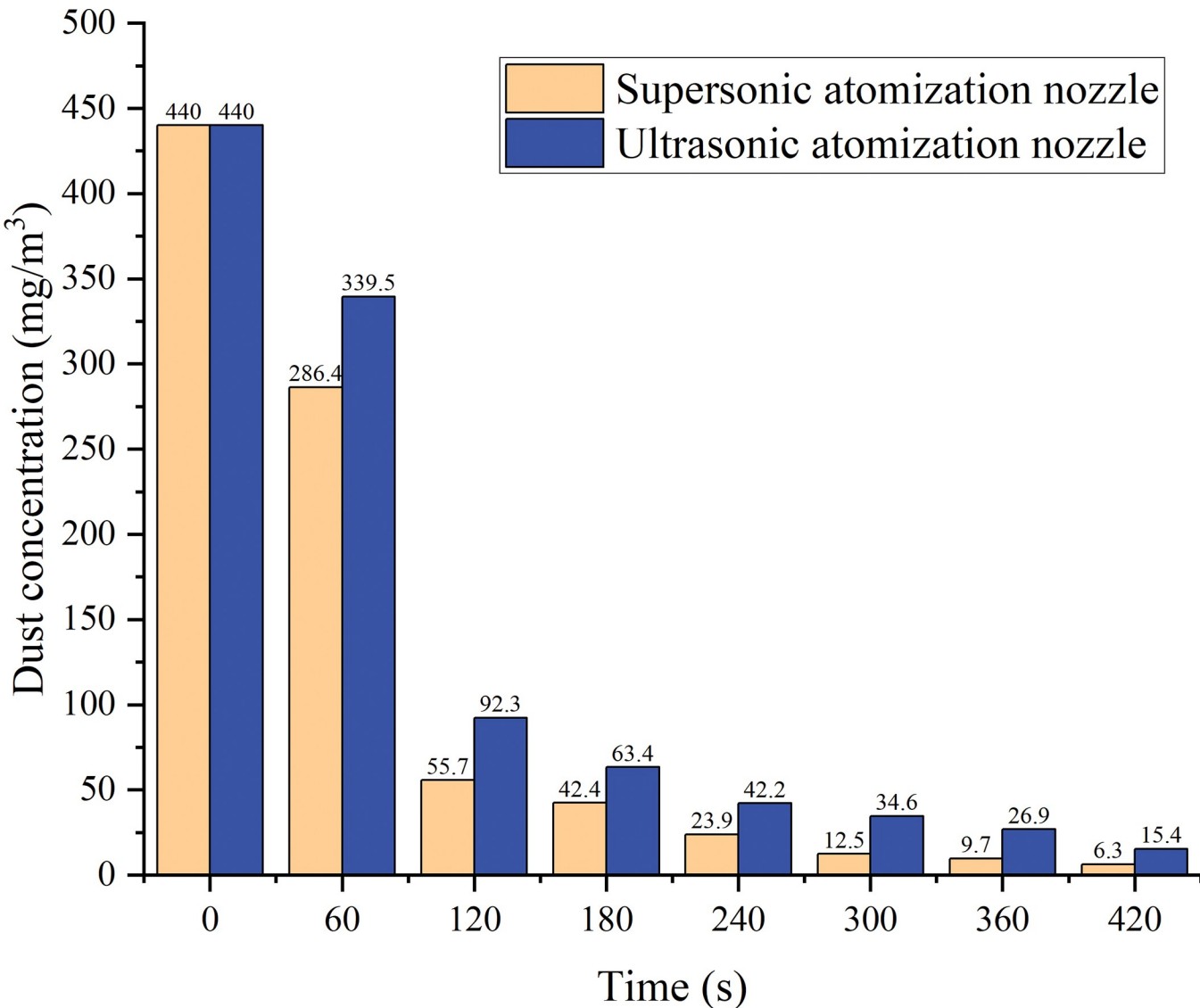

**Fig 13. Relationship between the dust-to-mist particle size ratio and the dust reduction efficiency.**

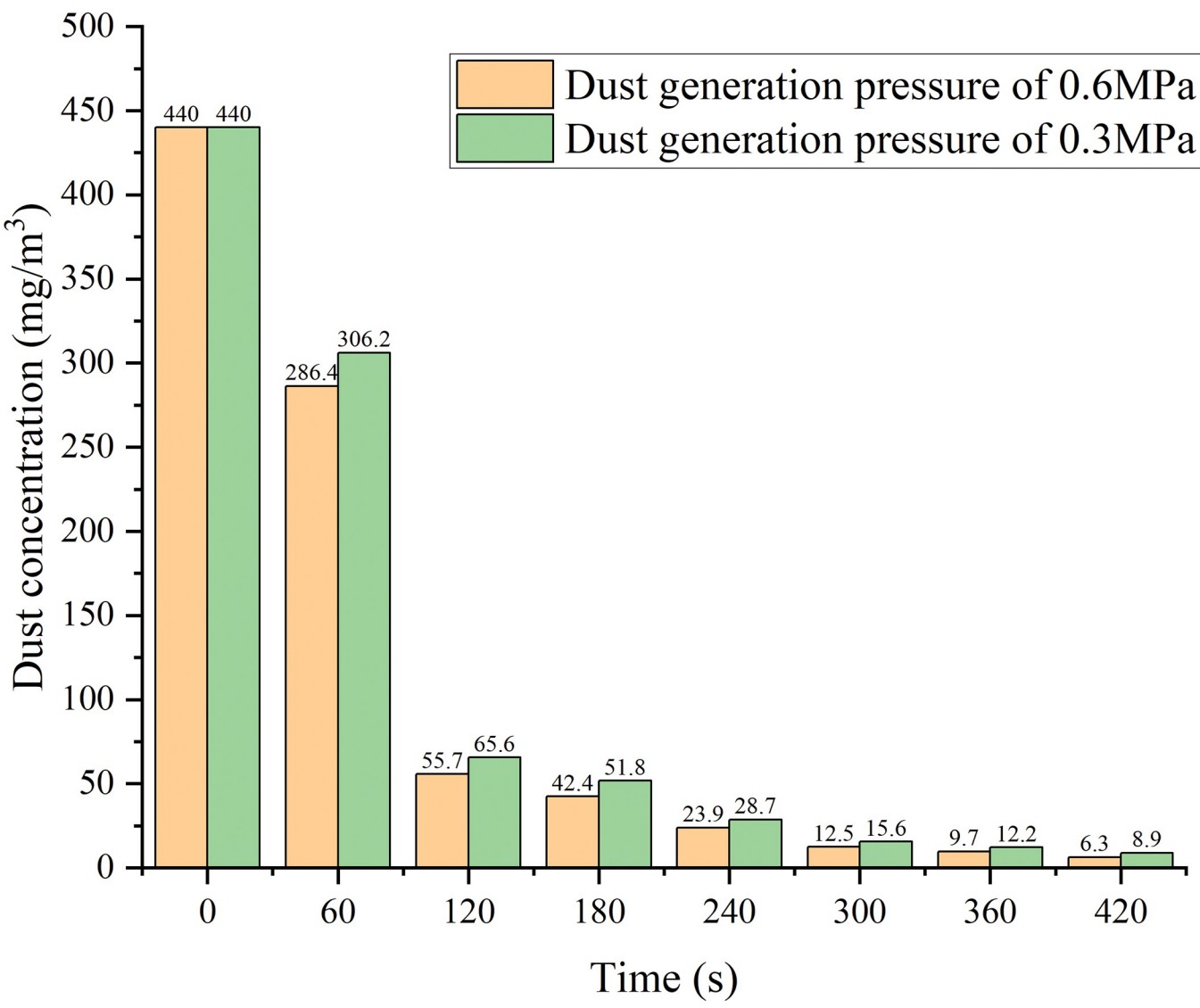

**Fig 14. Relationship between the dust concentration and dust reduction efficiency.**

to 0.3 MPa, the dust reduction efficiency changes from 98.47% to 97.98%. This result shows that the higher the relative velocity between the dust and the mist is, the higher the dust reduction efficiency is, all other conditions being equal. The conclusions obtained from the experimental data also validate the numerical simulation results.

## 6. Conclusion

A theoretical analysis was carried out to develop a numerical three-phase flow model of single-particle fog-dust collision-coupling in a high-speed airflow. The reliability of the numerical simulations was analysed in terms of the hydrophobic wetting and shielding phenomenon of dust, the streamline distribution for the fog-dust body during transient collisions and the change in the stress of the fog-dust interface during transient collisions.

1. The results of this study show that the optimal parameters for droplet collection of dust depend on the dust particle size, which provides information on the collision coupling

mechanism of micron-sized high-speed droplet and dust particle. At the beginning of a collision, the wettability of a dust particle is mainly influenced by the inertial force on the dust particle. When the inertial force on a dust particle is balanced against the air resistance, the fog-dust body relaxes to a state of uniform motion; the surface tension of the contact interface is the dominant influence factor for the wettability of the dust particles, and the stress distribution at the contact interface of the coupled particles reaches steady state at complete wetting.

2. At the micron-level, the collision process of high-speed droplet and dust particle is mainly affected by the fog-to-dust particle size ratio, relative velocity between the fog and dust particles, collision angle and contact angle at the solid–liquid surface; the fog-to-dust particle size ratio that produces the optimal wettability of the dust particles depends on the dust particle size. The results are $k_{PM20}$ = 1.5:1, $k_{PM10}$ = 3.5:1 and $k_{PM2.5}$ = 2:1, which shows that increasing the relative velocity between the fog and dust particles, $U$, results in an increase in the inertial forces on the dust particles and shorter relaxation times for the fog-dust body. The higher the dust-fog collision angle is, the larger the difference in the forces on the upper and lower sides of the dust particles is, the lower the wetting rate is, the smaller the contact angle at the solid-liquid interface during the collision of dust and fog particles is, and the better the wetting of the dust particles by the droplets is, holding all other conditions fixed.

3. The dust dispersity measured using different production procedures and in research studies is used to determine the optimal dust-to-mist particle size ratio for dust collection, which in turn is used to determine the optimal droplet dispersity. The droplet dispersity is a fundamental parameter for dust collection and provides theoretical support for the design of novel sprays, selection of multiple spray systems and optimization of dust removal solutions.

## Author Contributions

**Project administration:** Shaocheng Ge.

**Software:** Tian Zhang, Shaocheng Ge.

**Supervision:** Tian Zhang.

**Validation:** ShuaiShuai Ren, Mingxing Ma.

**Writing – original draft:** Deji Jing.

**Writing – review & editing:** Jichuang Ma.

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
