## [Decision Letter · Decision Letter 0]

3 May 2023

PONE-D-22-33806Numerical simulation of the fine kinetics of dust reduction using high-speed aerosolsPLOS ONE

Dear Dr. Zhang,

Thank you for submitting your manuscript to PLOS ONE. After careful consideration, we feel that it has merit but does not fully meet PLOS ONE’s publication criteria as it currently stands. Therefore, we invite you to submit a revised version of the manuscript that addresses the points raised during the review process.

We look forward to receiving your revised manuscript.

Kind regards,

Hongbing Ding, Ph.D.

Academic Editor

PLOS ONE

Journal Requirements:

jdj-National Natural Science Foundation of China Youth Fund Project (51704146)

jdj-Liaoning Provincial Natural Science Foundation (2020-MS-304)

jdj-Liaoning provincial funding for scientific research projects (LJK0323)

zt-China Postdoctoral Science Foundation (2022M11456)

NO authors have competing interests

Additional Editor Comments:

Thank you for submitting your manuscript to PLOS ONE. The reviewers recommend reconsideration of your paper following minor revision. I invite you to resubmit your manuscript after addressing all reviewer comments.

Reviewers' comments:

Reviewer's Responses to Questions

**Comments to the Author**

1. Is the manuscript technically sound, and do the data support the conclusions?

Reviewer #1: Yes

2. Has the statistical analysis been performed appropriately and rigorously? 

Reviewer #1: Yes

3. Have the authors made all data underlying the findings in their manuscript fully available?

Reviewer #1: Yes

4. Is the manuscript presented in an intelligible fashion and written in standard English?

Reviewer #1: Yes

5. Review Comments to the Author

Reviewer #1: The subject of the study is the single-particle collisional wetting process of micron-sized droplets on dust of different typical particle sizes in a high-speed airflow, which is investigated using a finite element-dynamic mesh modelling approach based on the three-phase flow theory. The reliability of the simulation is adequately given in terms of inter-particle forces, airflow streamlines and hydrophobic phenomena. A comparative study has been carried out to obtain the optimum wetting particle size ratio and collision angle for different typical particle sizes of dust. I think the results of the study can provide an important reference for the study of pneumatic spray capture of micron-sized particles. However, There are some minor problems with the article which should be slightly revised, I support the publication of this paper.

1.In this article, variables should be used in the correct format, page 10, paragraph 3 of the article.

2.Three typical particle sizes of dust particles are selected for study in the article, and the reasons for selecting the particle sizes PM2.5, PM10 and PM20 for study.

3.On page 19 of the article, heading 4.2, "At 1 µs, the relative motion velocity decreased to 1.5~2m/s." What does the relative motion velocity mean here.

4.In heading 4.3 on page 20 of the article, select four specific c values Please explain the reasons for the selection

5.The relative velocity between gas and solid is 30m/s to 100m/s. What is the basis for choosing this velocity.

6.In this paper, the collision between a single droplet and a single dust is studied. it is suggested to change " droplets and dust particle" to " droplet and dust particle"

7.The fonts in Fig. 13 and 14 should be adjusted to the same format.

8.What is the basis for the choice of the relative velocity between gas and solid of 30m/s to 100m/s set in the text?

6. PLOS authors have the option to publish the peer review history of their article (what does this mean?). If published, this will include your full peer review and any attached files.

Reviewer #1: No

---

## [Author Response · Author response to Decision Letter 0]

8 May 2023

Dear Editors and Reviewers:

Thank you for your letter and for the reviewers’ comments concerning our manuscript entitled “Numerical simulation of the fine kinetics of dust reduction using high-speed aerosols” (ID: PONE-D-22-33806). All comments are very important and they are great support for our scientific research work. We have studied comments carefully and have made correction (Revised portion are marked in red in the text). The main corrections in the paper and the responds to the reviewer’s comments are as flowing:

Reviewer 1

The subject of the study is the single-particle collisional wetting process of micron-sized droplets on dust of different typical particle sizes in a high-speed airflow, which is investigated using a finite element-dynamic mesh modelling approach based on the three-phase flow theory. The reliability of the simulation is adequately given in terms of inter-particle forces, airflow streamlines and hydrophobic phenomena. A comparative study has been carried out to obtain the optimum wetting particle size ratio and collision angle for different typical particle sizes of dust. I think the results of the study can provide an important reference for the study of pneumatic spray capture of micron-sized particles. However, There are some minor problems with the article which should be slightly revised, I support the publication of this paper.

Answers: Thank you very much for the excellent and professional revisions of our manuscript. We have studied the reviewer's comments carefully and have made revisions that have been marked in red in the revised manuscript.

Comment 1)In this article, variables should be used in the correct format, page 10, paragraph 3 of the article.

Answers: Thank you for your comments.I have changed the variables to the correct format .( Page 10)

Comment 2)Three typical particle sizes of dust particles are selected for study in the article, and the reasons for selecting the particle sizes PM2.5, PM10 and PM20 for study.

Answers: Thank you for your comments.The necessary boundary conditions for numerical simulations are described in the text. Three types of dust, PM2.5, PM10 and PM20, were chosen because they are more hazardous to humans in the micron range and the differences between the three are more pronounced all being representative of the dust.

Comment 3)On page 19 of the article, heading 4.2, "At 1 µs, the relative motion velocity decreased to 1.5~2m/s." What does the relative motion velocity mean here.

Answers: Thank you for your comments. The relative velocity here refers to the relative velocity between the dust particles and the water mist particles after their collision, i.e. the difference in velocity between the two particles.( Fig.9 has been modified accordingly)

Comment 4)In heading 4.3 on page 20 of the article, select four specific c values Please explain the reasons for the selection

Answers: Thank you for your comments. Four specific c values are selected for four collision angles, corresponding to 0°, 15°, 30° and 45°, and the range of collision angles is divided into four equal parts, and then four representative values are selected to better analyse the influence of collision angle on wettability and to obtain conclusions

Comment 5)The relative velocity between gas and solid is 30m/s to 100m/s. What is the basis for choosing this velocity.

Answers: Thank you for your comments.The main simulation in this paper is the flow field of high speed air movement, and the velocity of water mist at the exit of the supersonic atomisation nozzle can reach about 50m/s, so the relative velocity between gas and solid is selected from 30m/s to 100m/s, mainly to study the dust trapping of water mist particles moving at high speed. The effect of relative velocity on the dust trapping mechanism is effectively studied through simulation

Comment 6)In this paper, the collision between a single droplet and a single dust is studied. it is suggested to change " droplets and dust particle" to " droplet and dust particle"

Answers: Thank you for your comments.I have changed the variables to the correct format .

Comment 7)The fonts in Fig. 13 and 14 should be adjusted to the same format.

Answers: Thank you for your comments.I have changed the variables to the correct format .( Fig. 13 and 14)

Comment 8)In heading 4.3 on page 20 of the article, select four specific c values Please explain the reasons for the selection

Answers: Thank you for your comments.The authors mainly study the collision of dust and water mist airflow in high speed movement. Although the collision of engineering dust and water mist airflow cannot reach a high relative velocity at present, the authors find the conclusion of dust reduction efficiency in the flow field of high speed airflow movement through simulation research, which provides the theoretical basis for future research and development of dust reduction equipment, the greater the relative velocity the better the wetting effect and increase the dust capture efficiency. It is believed that future on-site dust reduction equipment can reach the collision situation of the high-speed airflow field, and the authors will also conduct future research on dust reduction equipment based on theoretical studies. Special thanks to the reviewers for the questions raised, and the authors will conduct deeper research on the relevant contents in future studies.

Thank you very much for your suggestion!

---

## [Editor Report · Decision Letter 1]

9 May 2023

Numerical simulation of the fine kinetics of dust reduction using high-speed aerosols

PONE-D-22-33806R1

Dear Dr. Zhang,

We’re pleased to inform you that your manuscript has been judged scientifically suitable for publication and will be formally accepted for publication once it meets all outstanding technical requirements.

Kind regards,

Hongbing Ding, Ph.D.

Academic Editor

PLOS ONE

Additional Editor Comments (optional):

The authors have done a good job in revising the manuscript. Now it can be accepted for publication in PLOS ONE.
---

## [Editor Report · Acceptance letter]

12 May 2023

PONE-D-22-33806R1 

Numerical simulation of the fine kinetics of dust reduction using high-speed aerosols 

Dear Dr. Zhang:

I'm pleased to inform you that your manuscript has been deemed suitable for publication in PLOS ONE. Congratulations! Your manuscript is now with our production department. 

Kind regards, 

on behalf of

Professor Hongbing Ding 

Academic Editor

PLOS ONE